# Prevalence and determinants of metabolic syndrome among long-shift healthcare professionals in primary hospitals of Central Gondar Zone, Northwest Ethiopia

Elias Chane◉*, Yilkal Amlaku, Amare Mekuanint, Abebaw Worede, Habtamu Wondifraw Baynes, Getnet Fetene

Department of Clinical Chemistry, School of Biomedical and Laboratory Sciences, College of Medicine and Health Sciences, University of Gondar, Gondar, Ethiopia

* eliaschane236@gmail.com, elias.chane@uog.edu.et

## Abstract

### Background

Metabolic syndrome (MetS) is a group of interrelated metabolic aberrations that significantly elevates the risk of poor cardiovascular outcomes and type 2 diabetes mellitus. Healthcare professionals, particularly those working long shifts, may have elevated risk due to the demanding nature of their work, irregular lifestyles, and associated stress. This study aimed to assess the prevalence and associated factors of MetS among healthcare professionals working long shifts in primary hospitals in the Central Gondar Zone, Northwest Ethiopia.

### Methods

An institutional-based cross-sectional study was conducted among a total of 271 healthcare professionals working in three primary hospitals (from September to December 2023). Study data were collected using structured questionairs, anthropometric measurements, and biochemical assessments. Five mililiters of fasting blood sample was collected from each participant; and serum lipid profile and glucose analyzed on Beckman Coulter DXC 700 AU chemistry analyzer. MetS was defined using the National Cholesterol Education Program Adult Treatment Panel III (NCEP ATP III) criteria. Independent ttest and one-way ANOVA were used for intra and inter group comparison; and Logistic regression model was fitted to identify factors associated with MetS, and adjusted odds ratios (AORs) with 95% confidence intervals (CIs) were reported to determine the strength of associations.

**Data availability statement:** All relevant data underlying the findings of this study are provided within the paper and its Supporting information files (S4 Dataset. Study dataset used for analysis).

**Funding:** The author(s) received no specific funding for this work.

**Competing interests:** The authors have declared that no competing interests exist.

**Abbreviations:** AORs, Adjusted odds ratios; BMI, Body Mass Index; X², Chi-square; CIs, Confidence intervals; COR, Crude Odds Ratio; DBP, Diastolic blood pressure; DDS, Dietary Diversity Score; ETB, Ethiopian Birr; FBG, Fasting blood glucose; HDL, HDL cholesterol; HC, Hip circumference; HPA, Hypothalamic-Pituitary-Adrenal axis; LDL, LDL cholesterol; MetS, Metabolic syndrome; MUAC, Mid Upper Arm Circumference; mg/dl, milligram per deciliter; ml, Milliliter; mmHg, Millimeters of Mercury; NCEP ATP III, National Cholesterol Education Program Adult Treatment Panel III; SBP, Systolic blood pressure; TC, Total cholesterol; TG, Triglycerides; T2DM, Type 2 diabetes mellitus; WC, Waist Circumference; WHR, Waist to Hip ratio; WHtR, Waist-to-Height Ratio.

## Results

The prevalence of MetS among healthcare professionals was 11.44% (95% CI 8.14–15.83). Dyslipidemias were observed to be the most common forms of metabolic derangement with 145 (53.51%) of study subjects having at least one lipid profile abnormality; whereas, hyperglycemias was the least common 27 (9.96%) form of metabolic abnormalities. Age ≥ 35 years (AOR = 6.75; 95% CI: 2.34–19.46), a family history of diabetes among first-degree relatives (AOR = 7.78; 95% CI: 2.57–23.53), and short sleep duration (<6 hours per day) (AOR = 7.78; 95% CI: 2.35–25.70) were significant factors associated with MetS ($p < 0.05$).

## Conclusion and recommondation

Metabolic syndrome is prevalent among healthcare professionals particularily those working long shifts; with age, family history of diabetes, and insufficient sleep identified as key risk factors. Hospital administrators and occupational health units should implement routine metabolic screening, optimized shift scheduling, and sleep hygiene support programs specifically for healthcare professionals working prolonged shifts, with particular attention to high-risk staff groups. Further workplace-based research is also needed to evaluate the effectiveness of these targeted interventions.

## Introduction

Metabolic syndrome (MetS) is a cluster of interrelated biochemical and functional abnormalities, including central obesity, high blood pressure, dyslipidemias, and impaired glucose metabolism and regulation [1,2]. These conditions together increase the risk of poor cardiovascular outcomes, type 2 diabetes mellitus (T2DM), and premature mortality [2,3]. Even though the scale varies widely among different populations, the prevalence of MetS is increasing dramatically over the past decades, driven by lifestyle changes and work-related factors [4,5]. According to the National Health and Nutrition Examination Survey, the prevalence of metabolic syndrome in adults increased from 25.3% to 34.2% in 2018 [6] similarly, a recent NHANES evidence suggests that metabolic syndrome remains a major public health burden, affecting nearly 40% adults in the 2023, with persistent age and racial/ethnic disparities and with a modest rise over time [7]; This burden may be even higher in low- and middle-income settings such as Ethiopia, where healthcare systems frequently face workforce shortages, limited resources, and high patient-to-provider ratios;for instance a 2023 meta-analysis estimating an overall prevalence of 32.4% across African populations, and some regions such as southern Africa showing even higher levels [8]. Conversely, the pooled prevalence of metabolic syndrome in Ethiopia particularly was found to be 30.0% according to a recent report from 25 studies [9].

Professionals working long shifts (≥40 hours/week), such as healthcare workers, are at a higer risk of acquiring MetS due to their work environment and characteristics [10,11]. Long work hours, night shifts, inconsistent eating patterns, and insufficient

physical exercise all contribute to poor metabolic profiles, which are widespread among healthcare workers [10,12]. Job related stress and sleep deprivation, further exacerbate this risk, leading to hormonal imbalances and metabolic dysfunction [5,13]. Lengthy working hours, usual night shifts, and irregular sleep patterns can disrupt circadian rhythms, impair metabolic homeostasis, and endorse sedentary lifestyles such as physical inactivity, poor dietary, and higher consumption of caffeine [5,13,14]. These reasons contribute to a heightened risk of developing MetS among individuals [4,5,14].

The pathophysiological mechanisms linking prolonged shifts and MetS involve multifaceted interactions between neuroendocrine, metabolic, and inflammatory processes [15]. Shift work and its associated sleep deprivation change the hypothalamic-pituitary-adrenal axis, leading to chronic elevation of the stress hormone cortisol [16,17]. Cortisol when persistently elevate causes several metabolic disturbances; such as central fat deposition, chronic insulin resistance, and even dyslipidemia. Higher cortisol levels also impair glucose metabolism [17,18]. Moreover, the altered sleep-wake cycles experienced during shift work interfere with the natural secretion of melatonin hormone, which is responsible for regulating the sleep cycle and various metabolic processes. Disrupted melatonin secretion not only affects sleep patterns but also impairs metabolic regulation, further exacerbating the risk of developing MetS [18,19].

Prolonged shift work has been associated to sympathetic nervous system activation; this persistent activity throughout lengthy shifts causes an increase in the production of stress hormones, such as catecholamines, which elevate blood pressure and contribute to the development of transient hypertension. In addition to these cardiovascular consequences, heightened sympathetic nervous system activity promotes a chronic inflammatory state within the body, defined by increased synthesis of pro-inflammatory cytokines (18, 19). This systemic inflammation further worsens insulin resistance and endothelial dysfunction which is a key contributors to MetS development [1,3,9]. Additionally, disrupted eating patterns contribute to metabolic derangements by impairing glucose and lipid metabolism [5].

In Ethiopia, healthcare professionals often face high workloads and extended shifts, yet limited research has examined the burden of MetS in this population [5,20]. Previous studies conducted in high-income countries have demonstrated a strong association between work-related factors, and the prevalence of MetS [5,14]. However, evidence from low- and middle-income countries, where healthcare professionals frequently work under high patient loads and limited resources, remains scarce. Therefore, this study aims to determine the prevalence and factors associated with MetS among healthcare professionals working long shifts in primary hospitals. We believe that understanding the burden and associated factors of MetS in this group is critical for designing targeted interventions to improve the health and well-being of healthcare professionals.

## Methodology

### Study design, and setting

A facility based cross sectional study was conducted among health professionals working at three primary hospitals located at Central Gondar Zone; from September 25 to December 29, 2023.

### Study population

The source populations for this study were all healthcare professionals at Central Gondar Zone and working longer shifts (≥40 hours/week). Healthcare professionals within the same hospitals often worked different duty hours depending on their professional role, ward assignment, emergency responsibilities, staffing availability, and rotating shift schedules, resulting in variability in shift duration exposure. The study populations were healthcare professionals at the selected primary hospitals who were "on duty" during the data collection period from September 25 to December 29, 2023.

### Eligibility criteria

**Inclusion criteria.** Eligible participants were healthcare professionals employed at primary hospitals in the Central Gondar Zone for at least 6 months, working a long-shift schedule (i.e., exceeding 40 hours per week), and this includes

those with irregular schedules such as rotating or night shifts. Participants were invited to take part voluntarily after providing written informed consent and, were included in the study. Healthcare professionals within the same hospital may work different hours depending on their professional role, ward assignment, emergency duties, staffing availability, and rotating schedules, resulting in variability in shift duration exposure.

**Exclusion criteria.** Healthcare professionals working part-time, as well as those who are pregnant or lactating at the time of data collection, were excluded from the study due to the potential impact of these conditions on metabolic parameters. Additionally, healthcare professionals with known pre-existing chronic conditions such as chronic kidney disease, uncontrolled cancer, or thyroid disorders were excluded, as these conditions may alter the metabolic profile. Participants who have undergone major surgery within the past six months or were recently diagnosed with serious medical conditions were also excluded due to their influence on metabolic parameters; these conditions were identified through self-reports from candidate participants. Lastly, individuals who did not complete the study protocol were removed from the study.

### Study variables

**Dependent variables.** Metabolic syndrome (Hyperglycemia, Hypertension, Central obesity and Dyslipidemia).

**Independent variables.** The study assessed a range of independent variables, including demographic factors such as age, gender, profession, marital status, average monthly income, educational level, and years of professional experience. Anthropometric and clinical characteristics were also measured, including body mass index (BMI), mid-upper arm circumference (MUAC), presence of comorbidities, and family history of diabetes and hypertension. Additionally, behavioral, lifestyle, and dietary-related factors were recorded, including sleep patterns and average sleep duration, habits of coffee drinking, alcohol consumption, smoking, khat chewing, level of physical activity, dietary diversity score (DDS), usual food source, consumption of fatty foods, type of oil or fat used, and intake of sugary foods and drinks. Finally, work characteristics and job-related factors were evaluated, including usual shift type, typical shift duration, weekly hours on duty, perceived stress level, job satisfaction, and workload perception.

### Sample size and sampling technique

The sample size is calculated using a single population proportion formula on Open-Epi software. The value of the dependent variable was taken from the previous study conducted in Addis Ababa, Ethiopia [21]; with assumptions of a 95% confidence level, 5% margin of error, an estimated prevalence of metabolic syndrome 17.9%, and 20% non-response rate; and we have reached a final sample size of 271.

A multistage sampling technique was employed to select study participants. In the first stage, three primary hospitals were randomly selected from the Central Gondar Zone. Simple random sampling was used to ensure each hospital had an equal chance of being included in the study. The selected hospitals serve comparable patient populations and healthcare workforce structures to other primary hospitals in the region, supporting the representativeness of the study sample. In the second stage, a complete list of healthcare professionals working long shifts was obtained from the human resource departments of the selected hospitals. Finally, individual participants were selected using simple random sampling from these strata to ensure representation and avoid selection bias. This approach ensured that the sample was representative of healthcare professionals working long shifts across the Central Gondar Zone primary hospitals. Efforts were made to maintain participant confidentiality and minimize disruptions to their work schedules during data collection.

### Operational definitions

• Alcohol Drinking Habit – Participants were asked about their frequency and quantity of alcohol any of the following alcoholic beverages such as beer, wine, spirits, or any other Ethiopian traditional alcoholic drinks; and was considered

"*Non-drinker*"- No alcohol consumption or less than one drink per month; "*Occasional drinker*" Consumes alcohol less than once per week; "*Regular drinker*" Consumes alcohol 1–7 times per week "*Heavy drinker*" Consumes alcohol more than 7 drinks per week or binge drinking (≥ 5 drinks in one sitting for men; ≥ 4 for women).

- BMI- was considered from weight and height measurement and categorized as < 18.5 kg/m$^2$ (*Underweight*), 18.5–24.9 kg/m$^2$ (*Normal*), 25.0–29.9 kg/m$^2$(*Overweight*), and ≥ 30 kg/m$^2$ (*Obese*) [22].

- Central Obesity- was considered WC > 90 cm (35.4 inches) for Men and WC > 80 cm (31.5 inches) for Women [23,24]

- Coffee Drinking Habit – frequency of coffee consumption over a typical week, regardless of quantity or preparation method; and was considered "*Rare/Never drinker*": Consumes coffee once or less than once per week or never, "*Weekly drinker*": Consumes coffee 1–6 days per week, and "*Daily drinker*" Consumes coffee every day (7 days per week).

- Dietary Diversity Score (DDS) – refers to the number of different food groups habitually consumed by an individual. Participants were asked to describe their typical dietary intake based on their routine eating patterns; and was considered "*Low dietary diversity DDS*" < 3 food groups, "*Moderate dietary diversity DDS*" 3–5 food groups, and "*High dietary diversity DDS*" ≥ 6 food groups.

- Dyslipidemia – serum Total cholesterol ≥ 200 mg/dL (5.2 mmol/L), or LDL cholesterol ≥ 130 mg/dL (3.4 mmol/L), or Triglycerides ≥ 150 mg/dL (1.7 mmol/L), or HDL cholesterol < 40 mg/dL (1.0 mmol/L) for men or < 50 mg/dL (1.3 mmol/L) for women [25].

- Family History of Diabetes/ Hypertension – was considered the presence of diabetes mellitus (Type 1 or Type 2) or hypertension in one or more first-degree relatives (parents, siblings, or children)

- Fast Food Utilization Habit- refers to habit of consuming fast food items such as fried ood, processed snacks, sandwiches and burgers, pizzas, sweetened beverages and desserts, packaged and ready-to-eat meals, any traditional Ethiopian quick foods.

- Hyperglycemia – was considered fasting blood glucose (FBG) ≥ 126 mg/dL (7.0 mmol/L) after an 8–10 hour fasting [26].

- Hypertension – Persistently elevated blood pressure, measured on at least two separate occasions; Systolic blood pressure (SBP) ≥ 140 mmHg, or Diastolic blood pressure (DBP) ≥ 90 mmHg [27].

- Khat Chewing Habit – use of khat (*Catha edulis*) leaves for recreational or social purposes; and was considered "*Khat Chewer*" and "*Non-chewer*" based on the participants response.

- Metabolic syndrome – was defined using "National Cholesterol Education Program (NCEP) Adult Treatment Panel III (ATP III)" guidelines as the presence of three or more of the following five criteria; Higher Abdominal Obesity (Waist Circumference), Elevated Triglycerides, Reduced HDL Cholesterol (Dyslipidemia) Or taking medication for reduced HDL cholesterol, Elevated Blood Pressure (Systolic Blood Pressure (SBP) ≥130 mmHg Or Diastolic Blood Pressure (DBP) ≥85 mmHg) or taking antihypertensive medication, Elevated Fasting Glucose (Hyperglycemia) or taking medication for elevated glucose [28].

- MUAC – was considered (*Severe malnutrition*) < 21 cm, (*Moderate Malnutrition*) 21–22.9 cm; and (*Normal*) ≥ 23 cm [29].

- Physical Activity – refers to any movement produced by skeletal muscles that requires energy expenditure and elevates heart rate. Adequate physical activity was defined according to WHO 2020 guidelines for adults, recommending at least 150 minutes of moderate-intensity aerobic activity per week, which roughly corresponds to 30 minutes per day on most

days. Participants' activity levels were categorized based on frequency as follows: "Inactive" Does not engage in any physical activity during the week; "Low" Engages in physical activity 1–2 days per week. "Moderate" Engages in physical activity 3–5 days per week and "High" Engages in physical activity 6 or more days per week [30].

- Sleep Duration- refers to the total number of hours an individual sleep in a 24-hour period, including naps. It was categorized into three levels "*Low Sleep Duration*"- Sleeping less than 6 hours per day (< 6 hours), "*Medium Sleep Duration*" – Sleeping between 6 and 8 hours per day (6–8 hours), and "*High Sleep Duration*" – Sleeping more than 8 hours per day (> 8 hours).

- Sleep Pattern – refers to the frequency and duration of interruptions during sleep; was categorized into three levels: "*Uninterrupted Sleep*"- defined as a continuous sleep period with no or minimal interruptions (e.g., waking up once or not at all) during the night. "*Occasionally Interrupted Sleep*"- defined as sleep interrupted 2–3 times during the night but with the ability to return to sleep relatively quickly after each interruption. and "*Frequently Interrupted Sleep*" – defined as sleep interrupted more than three times during the night, or characterized by significant difficulty in returning to sleep after interruptions.

- Waist Circumference (WC) – was considered "*Optimal*": < 90 cm (Men) or <80 cm (Women); and "Higher" ≥ 90 cm or ≥80 cm (Women) [23,24].

- Waist to Hip ratio (WHR)- was considered "*Lower risk*": < 0.9 (Men) or <0.85 (Women); and "*Higher risk*" ≥ 0.9 or ≥0.85 (Women) [23,24].

- Waist-to-Height Ratio (WHtR): was considered as "*Lower risk*" < 0.5 and "*Higher risk*" ≥ 0.5

**Data collection procedure**

The data collection process involved the use of pre-prepared structured questionnaires (S2 Text. **English version of the questionnaire**) which were translated into Amharic for convenience. The translated questionnaires were pre-tested through face-to-face interviews to ensure understanding based on the study objectives. Trained clinical nurses were selected as data collectors who screened participants for eligibility; participants who met the inclusion criteria were invited to participate voluntarily after being provided detailed information about the study and provided written and signed consent (S3 Text. **Participant information sheet**).

Weight was measured by means of an automatic digital weighing scale with an accuracy of ±0.1 kg, which was calibrated before measurement was taken ensuring participants wearing light clothing. Height was measured using digital stadiometer, with participants upright unshod on a flat area; The head was positioned in horizontal plane, and the height was recorded to the nearest 0.01 meter. Body mass index (BMI) was calculated as weight (kg) divided by the square of height (m²) and categorized using WHO classifications [22].

Waist circumference (WC) was measured using a non-stretchable measuring tape positioned at the midpoint between the lower margin of the last palpable rib and the top of the iliac crest. Hip circumference (HC) was measured at the widest part of the buttocks using the same tape, ensuring it remained parallel to the ground. Both WC and HC measurements were taken while participants stood upright with feet together, arms relaxed, and breathing normally and measurements were recorded to the nearest 0.1 cm. Mid-upper arm circumference (MUAC) was measured using a flexible, non-stretchable measuring tape on the participant's arm. The measurement was taken at the midpoint between the acromion process of the scapula and the olecranon process of the ulna, with the arm relaxed and hanging freely.

Resting blood pressure was measured using a sphygmomanometer and stethoscope from Omron, Japan, with readings taken from the upper arm of participants who had been seated for a minimum of 5 minutes. The cuff was placed securely on the upper arm of the non-dominant side, with the arm supported at heart level. Two readings were taken five

minutes apart, and the average of the two was used. In cases where the two readings differed significantly (≥10 mmHg), a third reading was taken, and the closest two readings were averaged.

## Laboratory analysis

Following face-to-face interviews with structured questions, fasting blood samples were collected from each participant after a minimum of consulted of 8-hour fast, processed according to SOP of the university of Gondar comprehensive specialized hospital's laboratory. 5 ml fasting venous blood sample of was collected from the cubital vein with a 21-gauge syringe and then transferred to a jell coated serum separator tube (SST); the tube was labeled with a unique ID number.

The blood was left to form a blood clot at room temperature for 45 minutes, then centrifuged with 5000 RPM for 3 minutes to separate cells from serum. Serum fasting glucose and lipid profiles was analyzed from serum using Beckman coulter dxc700 automated clinical chemistry analyzer; The DxC 700 AU clinical chemistry analyzer. Serum Glucose and Lipid profiles test results were reported as milligram per deciliter (mg/dl).

## Data quality control

Data collectors underwent comprehensive training prior to data collection period about the objective of the rsearch project, study participants data confidentiality, participant rights, consent procedures, interview techniques, and laboratory test protocols. A pre-test of the prepared questionnaire was conducted on a 5% sample size (14 healthcare professionals) of the target population to validate its accuracy and consistency. Sample analysis was carried out using a Beckman-Coulter DXC 700 chemistry analyzer, following quality control measures derived from feedback reports of the regional laboratory. The research team closely monitored the data collection and laboratory analysis process to maintain the data integrity and consistency.

## Statistical analysis and software

All participant data was first entered to Epi-data (v3.1) software, then transfer to STATA version 14 for statistical analysis. Descriptive statistics were used to characterize the data, while the Kolmogorov-Smirnov and plot tests were used to check the normal distribution of continuous numerical variables. Independent t-tests and one-way ANOVA were used for inter and intra group comparisons across different predictors, and a logistic regression model was fitted to identify associated factors, with statistical significance considered at $p < 0.05$, in the multivariable model, and $p < 0.25$ in bivariable model.

The independence and exclusivity of each variable were confirmed, and variables exhibiting multicollinearity were omitted from logistic analysis. Categorical independent variables were assessed using the chi-square ($X^2$) or Fisher's exact tests, with those meeting assumptions subjected to bivariate logistic regression analysis. Prior to logistic regression analysis, a rigorous variable screening process was implemented to select categorical independent variables. Each independent variable was first assessed using bivariate logistic regression. Variables with $p < 0.25$ were selected for multivariable analysis, following the standard recommendation for avoiding exclusion of potentially important variables in logistic regression [31]. Since we use STATA (v.14) it automatically omit multicolline variables; however, we have statistically evaluated using the variance inflation factor (VIF) and tolerance; variables with VIF > 10 or tolerance < 0.1 were considered collinear and excluded from multivariable models. Lastly, the logistic regression model was fitted and the Hosmer-Lemeshow test was analyzed to test the goodness of fit of the logistic regression model, yielding a p-value of 0.92.

## Ethical considerations

Ethical clearance was obtained from University of Gondar School of Biomedical and Laboratory Sciences research ethics committee. The study was conducted in accordance with the Declaration of Helsinki guidelines and regulations. Hard copy of ethically reviewed research proposal along with copy of ethical clearance was submitted to Amhara regional health

bureau and the the selected primary Hospitals. An official permission letter was obtained from the University of Gondar comprehensive and specialized hospital, Amhara regional health bureau and participant hospitals. Informed consent (**S3 Text. Participant information sheet.**) was also obtained from each study participant before the actual data collection. Participants were informed about the risks and benefits of the study; their right to withdraw at any time, confidentiality was maintained using codes, and their right to get their results for free.

## Results

### Socio-demographic characteristics

The socio-demographic characteristics of the 271 healthcare professionals included in the study are summarized in Table 1. The current study includes healthcare professionals aged between 23 and 59; with mean age of 33.91 (±8.33). The gender distribution revealed that 141 (52.03%) were male, while the remaining 130 (47.97%) were female professionlas. In terms of educational attainment, bachelor's degree was the most common qualification, representing 76.75% of the participants. Nurses constituted the largest professional group with 71 (26.20%), followed by Medical laboratory technology 64 (23.62%). Regarding years of experience, 120 (44.28%) had less than 3 years of experience in their field (Table 1).

**Table 1. Socio-demographic characteristics of Healthcare Professionals Working Long Shifts in Central Gondar Zone Primary Hospitals, Northwest Ethiopia, 2024 (n = 271, Gondar 2024).**

| Variable | Categories | Frequency | Percentage |
|---|---|---|---|
| **Age Group** | 23–34 | 167 | 61.62% |
| | 35 - 45 | 75 | 27.68% |
| | 45+ | 29 | 10.70% |
| **Gender** | Male | 141 | 52.03% |
| | Female | 130 | 47.97% |
| **Marital status** | Unmarried/Single | 118 | 43.54% |
| | Married | 139 | 51.29% |
| | Widowed/divorced | 14 | 4.17% |
| **Average Monthly Income** | < 5000 ETB | 19 | 7.01% |
| | 5000–10000 ETB | 137 | 50.55% |
| | 10000 + ETB | 115 | 42.44% |
| **Educational level** | Diploma | 19 | 7.01% |
| | First degree (BSc) | 208 | 76.75% |
| | Masters/MD/Higher | 44 | 16.24% |
| **Profession** | Nurse | 71 | 26.20% |
| | Medical doctor | 41 | 15.13% |
| | Lab technologist | 64 | 23.62% |
| | Pharmacist | 47 | 17.34% |
| | Other# | 48 | 17.71% |
| **Years of experience** | 1–3 years | 120 | 44.28% |
| | 4–10 years | 96 | 35.42% |
| | 11–20 years | 46 | 16.97% |
| | 20 + years | 9 | 3.32% |

\# Optometrist, Anesthesia, Radiologic technologist, midwives, Physical therapists and nutritionists.

ETB—Ethiopian Birr (1 ETB equals 0.008 USD (Ethiopian National Bank, December 2024)).

## Anthropometric and clinical characteristics

The mean BMI was 22.92 (±2.94) kg/m², 194 (71.59%) of participants fell into the normal body mass index category. The mean MUAC was 24.12 (±2.59) cm, reflecting well nutritional status. The mean WHR and WHtR were 0.87 (±0.04) and 0.43 (±0.06), respectively, with 176 (64.94%) of participants exceeding the recommended thresholds of 0.85 for WHR and 59 (21.77%) of participants had higher WHtR (> 0.5). The mean systolic and diastolic blood pressure values were 114.83 (±22.10) mmHg and 75.05 (±12.08) mmHg, respectively. 62 (22.88%) of participants reported a family history of diabetes, while 45 (16.61%) had a family history of hypertension (Table 2).

## Behavioral and Lifestyle characteristics

The behavioral and lifestyle characteristics of the study participants are summarized in Table 3. Majority 145(53.51%) of participants reported to have uninterrupted sleep. Regarding substance use 201 (74.17%) and 27 (9.96%) of participants reported not to consume alcohol and coffee respectively. Physical activity levels varied, with 168 (61.99%) of participants categorized as having low levels of physical activity; while 35 (12.92%) reported to be inactive. Daily fat-rich food utilization and the consumption of sugary foods or drinks were reported by 31 (11.44%) and 61 (22.51%), respectively *(Table 3)*.

## Working condition and job-related factors

The minimum shift duration was 4 hours per day, with 48 (17.71%) of participants reporting daily shifts longer than 12 hours. Similarly, the average hours on duty per week were 40 hours, with 41 (15.13%) participants reported to be engaged on their work for more than 55 hours per week, indicating potentially high workloads. Regarding stress levels, 91 (33.58%) of participants reported moderate to high job-related stress, reflecting the demanding nature of their roles. Despite this, 148 (54.61%) expressed satisfaction with their jobs, highlighting mixed perceptions of work-related factors. Additionally, 194 (71.59%) perceived their workload as heavy, which could contribute to stress and job dissatisfaction (Table 4).

**Table 2. Anthropometric and Clinical Characteristics of Healthcare Professionals Working Long Shifts in Central Gondar Zone Primary Hospitals, Northwest Ethiopia, 2024 (n = 271, Gondar 2024).**

| Variables | Categories | | Frequency | Percentage |
|---|---|---|---|---|
| BMI (kg/m²) | Under weight (< 18.5) | | 23 | 8.49% |
| | Normal (18.5–24.99) | | 194 | 71.59% |
| | Overweight and Obese (25+) | | 54 | 19.93% |
| MUAC (cm) | < 21 | | 30 | 11.07% |
| | 21–23 | | 54 | 19.93% |
| | 23 + | | 187 | 69.00% |
| Blood Pressure (mmHg) | SBP | < 140 mmhg | 229 | 84.50% |
| | | ≥140 mmhg | 42 | 15.50% |
| | DBP | < 90 mmhg | 213 | 78.60% |
| | | ≥ 90 mmhg | 58 | 21.40% |
| WHR | < 0.85 | | 95 | 35.06% |
| | ≥ 0.85 | | 176 | 64.94% |
| WHtR | < 0.5 | | 212 | 78.23% |
| | ≥ 0.5 | | 59 | 21.77% |
| Family history of Diabetics | None | | 209 | 77.12% |
| | Yes | | 62 | 22.88% |
| Family history of Hypertension | None | | 226 | 83.29% |
| | Yes | | 45 | 16.61% |

**Table 3. Behavioral and Lifestyle characteristics of Healthcare Professionals Working Long Shifts in Central Gondar Zone Primary Hospitals, Northwest Ethiopia, 2024 (n = 271, Gondar 2024).**

| Variable | Categories | Frequency | Percentage |
|---|---|---|---|
| **Sleep pattern** | Uninterrupted sleep | 145 | 53.51% |
| | Occasionally Interrupted | 49 | 18.08% |
| | Frequently Interrupted | 77 | 28.41% |
| **Smoking habit** | Non-smoker | 271 | 100.00% |
| | Smoker/ lives with smoker | – | – |
| **Alcohol drinking habit** | Non – drinker | 201 | 74.17% |
| | Occasional drinker | 47 | 17.34% |
| | Moderate drinker | 18 | 6.64% |
| | Heavy drinker | 5 | 1.85% |
| **Coffee drinking habit** | Rare/ Never | 27 | 9.96% |
| | Weekly/ Occasional | 186 | 68.63% |
| | Daily/ Regular | 58 | 21.40% |
| **Khat chewing habit** | Yes | 10 | 3.69% |
| | No | 261 | 96.31% |
| **Level of physical activity** | Inactive | 35 | 12.92% |
| | Low (1–2 days per week) | 168 | 61.99% |
| | Moderate (3–5 days per week) | 55 | 20.30% |
| | High (≥ 6 days per week) | 13 | 4.80% |
| **Dietary diversity score** | Low | 71 | 26.20% |
| | Medium | 111 | 40.96% |
| | High | 89 | 32.84% |
| **Usual food content** | Plant based Diet | 232 | 85.61% |
| | Animal based Diet | 39 | 14.39% |
| **Fast food utilization habit** | No | 240 | 88.56% |
| | Yes | 31 | 11.44% |
| **Sleep Duration** | Low (< 6 Hours) | 77 | 28.41% |
| | Medium (6–8 Hours) | 168 | 61.99% |
| | High (8 + Hours) | 26 | 9.59% |
| **Type of Oil/ fat** | Sunflower oil | 250 | 92.25% |
| | Soybean | 18 | 6.64% |
| | Olive oil | 3 | 1.11% |
| **Consumption of sugary foods/Drinks** | Rarely/ never | 177 | 65.31% |
| | 1 - 3 times per week | 18 | 6.64% |
| | 4–6 times per week | 15 | 5.54% |
| | Daily | 61 | 22.51% |

## Biochemical and anthropometric parameters

The mean waist circumference was 72.45 (±9.70) cm, with a mean WHR and WHtR of 0.875 (±0.045) and 0.439 (±0.064), respectively. The mean BMI was 22.92 (±2.94) kg/m². The average systolic and diastolic blood pressures were 114.83 (±22.10) mmHg and 75.05 (±12.08) mmHg, respectively (Table 5).

Independent ttest and one-way ANOVA test was fitted for selected variable categories. A statistically significant difference was observed in triglyceride (TG) levels between males and females, with females having a lower mean value

**Table 4. Working condition and job-related factors of Healthcare Professionals Working Long Shifts in Central Gondar Zone Primary Hospitals, Northwest Ethiopia, 2024 (n = 271, Gondar 2024).**

| Variable | Categories | Frequency | Percentage |
|---|---|---|---|
| **Usual shift Type** | Day | 59 | 21.77% |
| | Night | 14 | 5.17% |
| | Rotational | 198 | 73.06% |
| **Typical shift duration** | ≤ 8 Hours | 164 | 60.52% |
| | 8–12 hours | 59 | 21.77% |
| | 12 + Hours | 48 | 17.71% |
| **Hours on duty per week** | 40–48 Hours | 84 | 31.00% |
| | 49–54 Hours | 146 | 53.87% |
| | 55 & above Hours | 41 | 15.13% |
| **Stress level** | Low | 180 | 66.42% |
| | Moderate | 18 | 6.64% |
| | High | 15 | 5.54% |
| | Very high | 58 | 21.40% |
| **Job Satisfaction** | High Job Satisfaction | 148 | 54.61% |
| | Moderate Job Satisfaction | 81 | 29.89% |
| | Low Job Satisfaction | 42 | 15.50% |
| **Workload perception** | High Workload perception | 194 | 71.59% |
| | Moderate workload perception | 49 | 18.08% |
| | Low workload perception | 28 | 10.33% |

# Private Job, on maternal leave.

ETB – Ethiopian Birr (1 ETB equals 0.008 USD (Ethiopian National Bank, December 2024)).

**Table 5. Biochemical and Anthropometric parameters among of Healthcare Professionals Working Long Shifts in Central Gondar Zone Primary Hospitals, Northwest Ethiopia, 2024 (n = 271, Gondar 2024).**

| Variables | Mean | SD | Range |
|---|---|---|---|
| Waist Circumference (cm) | 72.45 | ±9.70 | 59–101 |
| WHR (cm/cm) | 0.875 | ±0.045 | 0.81–1.08 |
| WHtR (cm/cm) | 0.439 | ±0.064 | 0.32–0.64 |
| BMI (kg/m$^2$) | 22.92 | ±2.94 | 15.45–30.12 |
| SBP (mmHg) | 114.83 | ±22.10 | 90–200 |
| DBP (mmHg) | 75.05 | ±12.08 | 60–110 |
| FBS (mg/dl) | 90.42 | ±23.77 | 63–218 |
| Total Cholesterol (mg/dl) | 152.05 | ±29.54 | 99–236 |
| Triglyceride (mg/dl) | 84.97 | ±30.66 | 39–169 |
| HDL (mg/dl) | 46.02 | ±8.99 | 21–61 |
| LDL (mg/dl) | 82.07 | ±27.06 | 37–162 |

(mean difference = 8.64 mg/dL, Cohen's d = 0.28, p = 0.021); Similarly, individuals working exclusively on night shifts had a significantly lower mean systolic blood pressure than those working day shifts or rotational shifts (Cohen's d = 0.025, p = 0.017). Participants who reported having shorter sleep durations (<6 hours) due to their jobs had significantly lower BMI compared to those with optimal or longer sleep durations (Cohen's d = 0.121, p = 0.000). On the other hand, participants who reported having lower sleep durations (< 6 hours) observed to having statistically significant higher mean

values for diastolic blood pressure (DBP) (Cohen's d = 0.08, p = 0.000). and fating blood sugar (FBS) level (Cohen's d = 0.06, p = 0.000). Frequency of job-related sleep interruption is another factor observed to affect biochemical profile of study participants; those with frequently interrupted sleep observed to having statistically significant higher mean values of BMI (Cohen's d = 0.30, p = 0.000), both Systolic (Cohen's d = 0.14, p = 0.000), and diastolic blood pressure (Cohen's d = 0.14, p = 0.000), and fasting blood sugar level (Cohen's d = 0.112, p = 0.000), than participants with uninterrupted or occasionally interrupted sleep (Table 6).

### Magnitude of metabolic syndrome

The prevalence of metabolic syndrome using the National Cholesterol Education Program Adult Treatment Panel III (NCEP ATP III) diagnosing guidelines for metabolic syndrome was 11.44%. dyslipidemias were observed to be the most common forms of metabolic derangement among study participants with 145 (53.51%) of study subjects were exhibiting at least one lipid profile abnormality; contrary to this, hyperglycemias (FBS ≥ 126 mg/dl) was the least common metabolic derangement with only 27 (9.96%) of study subjects exhibit the condition (Table 7 & S1 Fig).

### Factors associated with metabolic syndrome

A logistic regression model was fitted to identify predictors associated with Metabolic syndrome. All variables with p ≤ 0.25 on bivariable logistic analysis was fitted in to multivariable logistic analysis. On multivariable logistic analysis age group, family history of diabetics and average sleeping duration showed a statistically significant association with MetS among study participants. The odds of having MetS was 6.75 times higher among participants with age 35 and above (AOR 6.75, 95% CI 2.34–19.46). Conversely, healthcare professionals with a family history of diabetics on in one or more first-degree relatives had 7.78 times higher odds for MetS (AOR = 7.78; 95% CI 2.57–23.53). On the other hand, Participants who reported having lower sleep duration (< 6 hours) per day had seven folds of higher odds for MetS syndrome (AOR = 7.78; 95% CI 2.35–25.70) (Table 8).

## Discussion

This study aimed to assess the prevalence of MetS and identify its factors associated with MetS among healthcare professionals working long shifts in primary hospitals of the Central Gondar Zone. Our findings showed} that the overall prevalence of MetS was 11.44% (95% CI 8.14–15.83) using the National Cholesterol Education Program Adult Treatment Panel III (NCEP ATP III) diagnosing guidelines, with higher age, family history of diabetes, and insufficient sleep duration significantly associated with increased risk. These results underscore the importance of occupational and lifestyle factors on metabolic health among hospital staff and set the stage for comparing our findings with previous studies in other settings.

A meta-analysis conducted to assess the prevalence of MetS among Ethiopian Population reported reported 34.89% prevalence among Ethiopians [32]. On this study, the prevalence of MetS among heathcare professionals exposed to longer working shifts was 11.44% which was lower than that of the general population, but it was similar to a previous study conducted in general working adults in Addis Ababa, Ethiopia reported a population prevalence of Mets 12.5% among working population using the NCEP ATP III guidelines [21]; This association among the reported magnitude of MetS can be clarified because the workers in the hospital were mostly younger and exhibited higher education level, as higher education was associated with Lower incidence of MetS [33,34].

The prevalence MetS observed in our study is consistent with findings reported from studies conducted among health care professionas in Taiwan 13.5% and 13.84% [35,36]. This similarity can be attributed to comparable occupational stressors and lifestyle factors. Healthcare professionals in both settings often work long shifts, encounter high job-related stress, and have irregular meal patterns, which are recognized risk factors for MetS [13,20]. Furthermore, sedentary behavior due to prolonged working hours, limited time for physical activity, and reliance on convenience foods are common characteristics among healthcare workers potentially contributing to a higher prevalence of MetS [13].

 

**Table 6. Comparison of Biochemical and Anthropometric parameters among Healthcare Professionals Working Long Shifts in Central Gondar Zone Primary Hospitals, Northwest Ethiopia, 2024 (n = 271, Gondar 2024).**

| Predictor | Outcome | | | | | | | | |
|---|---|---|---|---|---|---|---|---|---|
| Variables | Category | WHR | BMI | SBP | DBP | FBS | TC | TG | HDL | LDL |
| **Sex** | **Male** | 0.88 (±0.05) | 22.78 (±2.83) | 114.04 (±20.58) | 74.39 (±11.54) | 91.61 (±22.86) | 153.14 (±30.39) | 89.12 (±32.53) | 45.94 (±9.19) | 82.77 (±27.59) |
| | **Female** | 0.87 (±0.03) | 23.07 (±3.07) | 115.69 (±23.69) | 75.76 (±12.64) | 89.13 (±24.75) | 150.87 (±28.66) | 80.48 (±27.93) | 46.12 (±8.79) | 81.30 (26.55) |
| *p-value* | | *0.06* | *0.42* | *0.54* | *0.35* | *0.39* | *0.52* | *0.02\** | *0.86* | *0.65* |
| *Effect size (Cohen's d)* | | *0.23* | *−0.0* | *−0.07* | *−0.11* | *0.10* | *0.07* | *0.28* | *−0.01* | *0.05* |
| **Shift type** | **Day** | 0.877 (±0.04) | 23.28 (±2.99) | 120.93 (±28.12) | 77.45 (±14.69) | 94.89 (±31.12) | 157.72 (±32.41) | 84.79 (±32.29) | 44.74 (±9.13) | 86.59 (±31.85) |
| | **Night** | 0.885 (±0.06) | 23.13 (±3.35) | 106.78 (±16.36) | 70.35 (±9.89) | 89.92 (±20.44) | 153.42 (±35.28) | 88.42 (±40.21) | 47.0 (±8.83) | 79.50 (±29.32) |
| | **Rotational** | 0.874 (±0.04) | 22.80 (±2.91) | 113.58 (±20.05) | 74.67 (±11.25) | 89.12 (±21.31) | 150.26 (±28.13) | 84.78 (±29.56) | 46.34 (±8.97) | 80.90 (25.30) |
| *p-value* | | *0.64* | *0.52* | *0.03\** | *0.09* | *0.26* | *0.23* | *0.91* | *0.44* | *0.34* |
| *Effect size (eta squared (η²))* | | *0.003* | *0.004* | *0.025* | *0.017* | *0.009* | *0.010* | *0.0006* | *0.005* | *0.007* |
| **Average working hours per week** | **40–48 hrs.** | 0.883 (±0.05) | 22.75 (±2.82) | 114.16 (±20.22) | 74.34 (±11.48) | 90.84 (±23.71) | 152.41 (±29.65) | 88.34 (±32.02) | 44.79 (±9.70) | 82.50 (±25.53) |
| | **48–54 hrs.** | 0.870 (±0.04) | 23.08 (±2.93) | 115.47 (±22.35) | 74.86 (±11.75) | 89.54 (±23.52) | 152.84 (±29.14) | 83.02 (±29.58) | 47.04 (±8.53) | 82.70 (±27.76) |
| | **≥ 55 hrs.** | 0.879 (±0.05) | 22.70 (±3.28) | 113.90 (±25.21) | 77.19 (±14.31) | 92.70 (±25.21) | 148.51 (±31.16) | 85.02 (±31.78) | 44.92 (±8.83) | 78.92 (±28.00) |
| *p-value* | | *0.10* | *0.63* | *0.87* | *0.44* | *0.74* | *0.70* | *0.45* | *0.13* | *0.72* |
| *Effect size (eta squared (η²))* | | *0.016* | *0.003* | *0.001* | *0.005* | *0.002* | *0.002* | *0.005* | *0.015* | *0.002* |
| **Sleep duration** | **Low Sleep Duration** | 0.88 (±0.04) | 22.18 (±2.57) | 112.74 (±27.39) | 80.51 (±13.94) | 99.63 (±33.79) | 158.67 (±32.74) | 89.81 (±35.02) | 42.55 (±9.39) | 88.85 (±30.92) |
| | **Medium Sleep-Duration** | 0.870 (±0.04) | 23.04 (±2.67) | 108.86 (±15.61) | 72.67 (±10.30) | 86.19 (±16.17) | 149.07 (±28.85) | 83.31 (±29.72) | 47.41 (±8.42) | 79.10 (±25.10) |
| | **Higher Sleep Duration** | 0.88 (±0.04) | 24.50 (±3.20) | 114.61 (±24.53) | 74.23 (±12.05) | 90.46 (±22.48) | 151.69 (±20.21) | 81.38 (±20.45) | 47.34 (±8.86) | 81.11 (±24.23) |
| *p-value* | | *0.03\** | *0.000\** | *0.000\** | *0.000\** | *0.000\** | *0.06* | *0.25* | *0.0003* | *0.03* |
| *Effect size (eta squared (η²))* | | *0.022* | *0.121* | *0.145* | *0.08* | *0.06* | *0.02* | *0.010* | *0.05* | *0.025* |
| **Sleep Interruption** | **Uninterrupted sleep** | 0.872 (±0.04) | 21.69 (±2.36) | 109.31 (±16.15) | 72.31 (±10.49) | 85.62 (±15.78) | 150.47 (±28.64) | 83.53 (±30.10) | 47.40 (±8.26) | 81.28 (±26.17) |
| | **Occasionally Interrupted** | 0.864 (±0.03) | 22.59 (±2.48) | 110.00 (±15.13) | 71.83 (±9.50) | 84.75 (±11.18) | 149.04 (±27.57) | 82.10 (±28.64) | 48.10 (±8.45) | 76.08 (±24.84) |
| | **Frequently Interrupted** | 0.887 (±0.04) | 25.45 (±2.63) | 128.31 (±28.90) | 82.27 (±13.39) | 103.06 (±35.09) | 156.94 (±32.10) | 89.51 (±32.79) | 42.11 (±9.52) | 87.35 (±29.34) |
| *p-value* | | *0.01\** | *0.000\** | *0.000\** | *0.000\** | *0.000\** | *0.21* | *0.29* | *0.000\** | *0.06* |
| *Effect size (eta squared (η²))* | | *0.032* | *0.30* | *0.14* | *0.14* | *0.112* | *0.011* | *0.009* | *0.076* | *0.020* |

*Statistically significant by independent t-test/ one-way ANOVA is used for comparison

However, the Prevalnece of MetS in the current study is lower than reported studies conducted from, Brazil 24.4%, 38.1% and 28.0% [37–39]. Mexico 38.7% [40], Canada 22.0% [41], Sri Lanka 36.6% [42], Japan 17.6% [43] and Belgium 23.8% [44]. This variation could be attributed to differences in diagnostic criteria, which can lead to variations in prevalence rates. Our study specifically utilized NCEP ATP III diagnosing guidelines; simlilarly, population demographics, such

**Table 7. Magnitude of Metabolic Abnormalities among Healthcare Professionals Working Long Shifts in Central Gondar Zone Primary Hospitals, Northwest Ethiopia, 2024 (n = 271, Gondar 2024).**

| Metabolic Derangement | Magnitude | |
|---|---|---|
| | n (%) | CI (95%) |
| *Hypertension* | 68 (25.09%) | 20.26–30.63 |
| *Hyperglycemia* | 27 (9.96%) | 6.90–14.16 |
| *Central Obesity (WC)* | 30 (11.07%) | 7.83–15.52 |
| *Increased TC* | 24 (8.86%) | 5.91–12.90 |
| *Increased TG* | 16 (5.90%) | 3.63–9.44 |
| *Decreased HDL* | 127 (46.86%) | 40.95–52.85 |
| *Increased LDL* | 25 (9.23%) | 6.29–13.32 |
| *Dyslipidemias* | 145 (53.51%) | 47.50–59.40 |
| *Mets* | 31 (11.44%) | 8.14–15.83 |

as younger age, lower genetic predisposition, and different sex distributions, may also contribute to the lower prevalence in our study. Lifestyle differences, such as traditional high-fiber diets among Ethiopians, contrast with the carbohydrate-rich and processed food-heavy diets in some of the higher-prevalence countries [45]. On the other hand, the Prevalnece of MetS on our results higher than previous reports from India 3.57% [46] and France 3.87% [47], potentially due to differences in demographic characteristics, settings, or methodological approaches.

On the current study female healthcare professionasl observed to have lower had A statistically significant lower tri-glyceride levels than females, with females having a lower mean value 89.12 (±32.53) vs 80.48 (±27.93) (p values 0.02) this may be attributed to biological, hormonal, and lifestyle factors. Hormones such as estrogen are known to play a pro-tective role in lipid metabolism, contributing to lower triglyceride levels in females, especially premenopausal women [48]. On the other hand, individuals working exclusively night shifts had a statistically significant (p = 0.03) lower mean systolic blood pressure compared to those working day shifts or rotational shifts. The mean systolic blood pressure values were 120.93 (±28.12) for day shift workers, 106.78 (±16.36) for night shift workers, and 113.58 (±20.05) for those on rotational shifts. Night shift workers may experience reduced exposure to workplace-related stressors that are more common during the day, such as high workloads and frequent interruptions. Additionally, variations in circadian rhythms could also be contribute, as blood pressure naturally fluctuates throughout the sleep-wake cycle, potentially leading to lower readings during the night [49,50].

Healthcare professionals who reported having shorter sleep durations (<6 hours) due to their duty time had statistically significant lower BMI 22.18 (±2.68) compared to those with optimal 23.43 (±3.07) or longer sleep durations 24.93 (±2.59) (p<0.0001). This finding may reflect short-term physiological and behavioral effects of prolonged wakefulness, including increased energy expenditure, higher occupation-related activity during extended shifts, and reduced opportunities for meal consumption during busy duty hours. [51,52]. However, evidence from longitudinal studies suggests that chronic sleep deprivation contributes to metabolic dysregulation, hormonal imbalance, and weight gain over time [52,53], There-fore, the lower BMI observed in our study should be interpreted as a possible acute or short-term phenomenon rather than a protective long-term metabolic effect.

Healthcare professionals who reported having longer sleep durations (8 + hours) were observed to having statistically significant higher BMI, DBP, and FBS (p<0.0001) level. This finding may be explained by underlying health conditions associated with longer sleep, such as fatigue, poor sleep quality, or metabolic disturbances, which are often linked to higher blood pressure and body glucose dysregulation [54,55]. Additionally, longer sleep durations might be associated with reduced physical activity during waking hours, contributing to unfavorable metabolic outcomes.

**Table 8. Factors Associated with Metabolic syndrome among Healthcare Professionals Working Long Shifts in Central Gondar Zone Primary Hospitals, Northwest Ethiopia, 2024 (n = 271, Gondar 2024).**

| Variables | Categories | MetS | | COR (95% CI) | AOR (95% CI) | p-value |
|---|---|---|---|---|---|---|
| | | **Yes** | **No** | | | |
| **Age Group** | < 35 years | 10 (5.99%) | 157 (94.01%) | Ref | Ref | |
| | ≥ 35 years | 21 (20.19%) | 83 (79.81%) | 3.97 (1.78–8.82) | 6.75 (2.34–19.46) | 0.000* |
| **Average working hour per week** | 40–48 Hours | 10 (11.90%) | 74 (88.10%) | Ref | Ref | |
| | 49–54 Hours | 13 (8.90%) | 133 (91.10%) | 0.72 (0.30–1.72) | 0.58 (0.19–1.77) | 0.34 |
| | ≥ 55 Hours | 8 (19.51%) | 33 (80.49%) | 1.79 (0.64–4.95) | 2.33 (0.55–9.79) | 0.24 |
| **WHtR** | < 0.5 | 20 (9.43%) | 192 (90.57%) | Ref | Ref | |
| | ≥ 0.5 | 11 (18.64%) | 48 (81.36%) | 2.20 (0.98–4.90) | 2.50 (0.84–7.38) | 0.09 |
| **Family History of Diabetes** | None | 9 (4.31%) | 200 (95.69%) | Ref | Ref | |
| | Yes | 22 (35.48%) | 40 (64.52%) | 12.22 (5.24–28.49) | 7.78 (2.57–23.53) | 0.000* |
| **Coffee drinking habit** | Rare/ None | 5 (18.52%) | 22 (81.48%) | Ref | Ref | |
| | Weekly | 18 (9.68) | 168 (90.32%) | 0.47 (0.15–1.39) | 0.21 (0.04–1.04) | 0.05 |
| | Daily/ Regular | 8 (13.79%) | 50 (86.21%) | 0.70 (0.20–2.39) | 0.28 (0.05–1.64) | 0.16 |
| **Fast food consumption** | None | 30 (12.50%) | 210 (87.50%) | Ref | Ref | |
| | Yes | 1 (3.23%) | 30 (96.77%) | 0.23 (0.03–1.77) | 0.31 (0.03–3.26) | 0.33 |
| **Average sleeping Duration** | Low (< 6 hrs.) | 24 (31.17%) | 53 (68.83%) | 14.76 (5.36–40.61) | 7.78 (2.35–25.70) | 0.001* |
| | Medium (6–8 hrs.) | 5 (2.98%) | 163 (97.02%) | Ref | Ref | |
| | Higher (8 + hrs.) | 2 (7.69%) | 24 (92.31%) | 2.71 (0.49–14.79) | 1.51 (0.17–13.15) | 0.70 |

*Statistically significant in Multivariable logistic regression.

Similarly, frequency of job-related sleep interruption is another factor observed to affect biochemical profile of study participants; those with frequently interrupted sleep observed to having statistically significant higher mean values of BMI, blood pressure and FBS level (p < 0.0001) than participants with uninterrupted or occasionally interrupted sleep. Sleep interruptions can lead to chronic stress, disruption of the circadian rhythm, and activation of the hypothalamic-pituitary-adrenal (HPA) axis, all of which contribute to adverse metabolic effects such as weight gain, elevated blood pressure, and impaired glucose regulation [56–58]. Furthermore, poor sleep quality can reduce insulin sensitivity and exacerbate systemic inflammation, further increasing the risk of metabolic abnormalities [58,59].

A logistic regression model was fitted to determine factors independently associated with metabolic syndrome among the study participants. The multivariable analysis identified three significant predictors: age group, family history of diabetes, and average sleeping duration. Healthcare professionals aged 35 years and above were found to have significantly higher odds of developing MetS, with an adjusted odds ratio (AOR) of 6.75 (95% CI: 2.34–19.46). This finding aligns with a recent report from Spain [60], Sri Lanka [42] and Japan [43] which reported advanced age associated with higher rski for MetS, this might be due to since aging universally leads to physiological changes, such as increased insulin resistance, central adiposity, and altered lipid metabolism, which elevate the risk of metabolic syndrome [61].

On the present study family history of diabetes among first-degree relatives also emerged as a significant predictor, with participants in this group exhibiting 7.78 times higher odds of metabolic syndrome (AOR = 7.78, 95% CI: 2.57–23.53). Individuals with a family history of diabetes are more likely to inherit genetic variants that contribute to insulin resistance, impaired glucose metabolism, and lipid abnormalities, all of which are key components of metabolic syndrome [62]. Additionally, shared environmental factors, such as dietary habits, physical activity levels, and lifestyle behaviors, within families further exacerbate the risk. Studies have consistently shown that a positive family history of diabetes is associated with higher prevalence rates of obesity, dyslipidemia, and hypertension, which are central features of metabolic syndrome [62,63]. Moreover, insufficient sleep duration (<6 hours per day) was associated with a sevenfold increase in the likelihood of metabolic syndrome (AOR = 7.78, 95% CI: 2.35–25.70). This finding aligns with existing evidence linking poor sleep hygiene to metabolic dysregulation, possibly due to its effects on hormonal balance, appetite regulation, and systemic inflammation [50–52].

## Conclusion and recommendations

The current study revelied that, the prevalence of MetS among healthcare professionals working long shifts in primary hospitals in the Central Gondar Zone was found to be 11.44% using NCEP ATP III guidelines. Higher age, family history of diabetes, and insufficient sleep duration were identified as key factors associated with MetS. To the best of our knowledge, this is the first study in Ethiopia to specifically examine metabolic syndrome among healthcare professionals working prolonged shifts, highlighting an important yet under-recognized occupational health issue. While the prevalence of MetS among healthcare workers in our study is lower than that of the general population in Ethiopia and other countries with higher rates, it still underscores the importance of addressing work-related and lifestyle factors to mitigate the risk of MetS.

To effectively address MetS among healthcare professionals working long shifts, a multi-pronged approach is crucial. This includes implementing comprehensive educational programs that emphasize healthy lifestyles, including nutrition, physical activity, sleep hygiene, and stress management. Developing a supportive workplace environment is equally important, which can be achieved by providing access to on-site fitness facilities, offering healthy food options, and encouraging regular breaks. Prioritizing sleep health is essential, and this can be addressed through the implementation of sleep management programs for shift workers, including education on sleep hygiene and strategies to improve sleep quality. Regular health screenings, particularly for those aged 35+ or with a family history of diabetes, are crucial to enable early detection and intervention. Lastly, continuing research is necessary to investigate the long-term impact of shift work on MetS and evaluate the effectiveness of interventions designed to reduce MetS prevalence in this population.

## Strength and limitation of the study

### Strength

To the best of our searching effort this study is the first study done in Ethiopia to assess the association between MetS and long shift working among health care professionals, thus it will provide a valuable insights into the prevalence and associated factors of metabolic syndrome among healthcare professionals working long shifts by specifically focusing on a definite group of individuals who are likely to experience unique stressors and health risks due to their work environment,

the study offers a deeper understanding of how long shifts may contribute to MetS. The use of an institutional-based cross-sectional design ensures that the data is collected from a real-world setting, increasing the study's relevance and applicability. The study also incorporates both clinical measurements (such as blood pressure, lipid profiles, and body measurements) and questionnaire data, providing a comprehensive view of the factors that contribute to MetS. Additionally, the study's findings could have significant implications for designing targeted health interventions to address the specific health risks faced by healthcare workers.

### Limitations

As a cross-sectional design, it can only establish associations rather than causal relationships, which limits our ability to draw definitive conclusions about the direct impact of long shifts on metabolic syndrome. Additionally, the study was conducted within a specific institutional setting, which may limit its generalizability to other healthcare environments with different work practices, populations, or regional factors. Another limitation is the reliance on self-reported data regarding behavioral and lifestyle characteristics, which were aligned with the expectation of maintaining good health. However, such self-reporting may not accurately reflect actual behaviors, as participants may have provided responses influenced by social desirability bias or the expectations about health practices in the healthcare profession. This could result in overestimation of healthy behaviors. Some subgroup analyses, particularly among exclusive night-shift workers, involved relatively small sample sizes, which may limit the precision and generalizability of those findings. Future studies should consider using objective monitoring methods, such as actigraphy for sleep assessment, wearable devices for physical activity tracking, and larger multicenter samples to improve the robustness and external validity of the findings. Furthermore, the study does not account for potential confounding factors, such as lifestyle habits outside the workplace, which may also contribute to metabolic syndrome. Finally, the study did not include hormonal testing (e.g., melatonin, ghrelin, leptin), which could provide valuable insight into the physiological mechanisms underlying metabolic syndrome in healthcare professionals working long shifts. These factors should be considered when interpreting the findings of this study.

### Scope statement

This study assesses the prevalence and factors associated with metabolic syndrome among healthcare professionals working long shifts in primary hospitals. The study is an institutional-based cross-sectional analysis that highlights key anthropometric, clinical, lifestyle, and behavioral risk factors influencing the development of metabolic syndrome. The findings address critical public health concerns related to occupational health, particularly for healthcare workers exposed to extended work hours and their associated stressors. By examining a vulnerable and often overlooked group, the study aligns with the mission of PLOS ONE to promote innovative research addressing health challenges within specific populations. The study's focus on preventive measures and policy implications further reinforces its relevance to the journal's scope and its audience of researchers and practitioners dedicated to improving population health.

### Supporting information

**S1 Fig. Magnitude of metabolic abnormalities among healthcare professionals working long shifts in Central Gondar Zone primary hospitals, Northwest Ethiopia, 2024.**
(PDF)

**S2 Text. English version of the questionnaire.**
(DOCX)

**S3 Text. Participant information sheet.**
(DOCX)

**S4 Dataset. Study dataset used for analysis.**
(XLS)

**S5 Checklist. STROBE checklist for cross-sectional studies.**
(DOCX)

## Acknowledgments

We would like to express our sincere gratitude to the healthcare professionals who participated in this study for their time and valuable contributions. We also extend our appreciation to the Central Gondar Zone health authorities for their support and cooperation during the research process. Lastly, we acknowledge the University of Gondar for providing the resources and ethical approval that made this study possible.
  Consent for publication
  All participants provided written informed consent to publish this study.
  Consent to participate
  Each participant was informed in detail and his/her consent was obtained before the data collection.
  Publisher's Note
  Open access.

## Author contributions

**Conceptualization:** Elias Chane, Yilkal Amlaku, Getnet Fetene.

**Data curation:** Elias Chane, Yilkal Amlaku, Abebaw Worede, Habtamu Wondifraw Baynes, Getnet Fetene.

**Formal analysis:** Elias Chane, Yilkal Amlaku, Amare Mekuanint, Abebaw Worede, Habtamu Wondifraw Baynes, Getnet Fetene.

**Funding acquisition:** Elias Chane, Habtamu Wondifraw Baynes, Getnet Fetene.

**Investigation:** Elias Chane, Yilkal Amlaku, Amare Mekuanint, Abebaw Worede, Habtamu Wondifraw Baynes, Getnet Fetene.

**Methodology:** Elias Chane, Yilkal Amlaku, Amare Mekuanint, Abebaw Worede, Habtamu Wondifraw Baynes, Getnet Fetene.

**Project administration:** Elias Chane, Amare Mekuanint, Abebaw Worede, Getnet Fetene.

**Resources:** Elias Chane, Yilkal Amlaku, Abebaw Worede, Getnet Fetene.

**Software:** Elias Chane, Getnet Fetene.

**Supervision:** Elias Chane, Amare Mekuanint, Habtamu Wondifraw Baynes, Getnet Fetene.

**Validation:** Elias Chane, Yilkal Amlaku.

**Visualization:** Elias Chane.

**Writing – original draft:** Elias Chane, Yilkal Amlaku.

**Writing – review & editing:** Elias Chane, Habtamu Wondifraw Baynes, Getnet Fetene.

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
