## [Decision Letter · Decision Letter 0]

4 Mar 2026

PONE-D-25-35581Prevalence and Determinants of Metabolic Syndrome Among Healthcare Professionals Working Long Shifts in Central Gondar Zone Primary Hospitals, Northwest Ethiopia: An Institutional-Based Cross-Sectional StudyPLOS One

Dear Dr. Chane,

Thank you for submitting your manuscript to PLOS ONE. After careful consideration, we feel that it has merit but does not fully meet PLOS ONE’s publication criteria as it currently stands. Therefore, we invite you to submit a revised version of the manuscript that addresses the points raised during the review process.

We look forward to receiving your revised manuscript.

Kind regards,

Edmund Tetteh Nartey

Academic Editor

PLOS One

Journal Requirements:

3. In the online submission form, you indicated that most of data generated or analyzed during this study are included in this published article; and Addition research data set associated with a paper is available, can be accessed in contact with the corresponding author.

Reviewers' comments:

Reviewer's Responses to Questions

**Comments to the Author**

1. Is the manuscript technically sound, and do the data support the conclusions?

Reviewer #1: Partly

Reviewer #2: Partly

2. Has the statistical analysis been performed appropriately and rigorously? 

Reviewer #1: I Don't Know

Reviewer #2: Yes

3. Have the authors made all data underlying the findings in their manuscript fully available?

Reviewer #1: Yes

Reviewer #2: Yes

4. Is the manuscript presented in an intelligible fashion and written in standard English?

Reviewer #1: Yes

Reviewer #2: Yes

5. Review Comments to the Author

Reviewer #1: General Comments: This study is well-structured and addresses a relevant topic, providing meaningful insights that can inform both research and practice in the field. Below, you wil find specific comments and suggestions to further improve the manuscript

Specific Comments:

Point1: In the intro section, you mention NHANES data up to 2018, which is fine, but since newer reports exist, the introduction would benefit from including the latest trends, at least briefly. It shows the reader that you are fully up to date.

Point2: The shift from general mechanisms to the Ethiopian setting feels a bit abrupt!! You might add one sentence explaining why low- and middle-income settings are particularly vulnerable (e.g.,, understaffing, resource constraints, high patient load).

Point3: Your description of the study setting and period is clear. However, you may want to justify why these three specific hospitals were selected and whether their patient/worker profile differs from other facilities in the region. This would help clarify representativeness and potential site-selection bias.

Point4: Some exclusion criteria (e.g., “serious conditions,” “recent major surgery”) are very broad and open to subjective interpretation. Please clarify how these were assessed (self-report, clinician confirmation, medical records).

Point5: Anthropometry and blood pressure measurement require clarification:

– were tools calibrated daily?

– were measurements taken twice and averaged?

– same arm used for BP?

Point6: If DDS was based on a single day, this is highly vulnerable to day-to-day variation. Clarify whether it was one recall or multiple days, and if only one day was used, I think is very important to acknowledge this limitation.

Point7: The definition (“20–30 min daily”) is not aligned with international norms (e.g., WHO). This may misclassify participants and affect associations. You should justify or revise the operational definition.

Point8: It would be useful to specify:

– how multicollinearity was assessed and handled

– why p<0.25 was used for bivariable screening (cite reference)

– how missing data were treated.

Point7: The results are well-presented and comprehensive, which is great. However, some anthropometric and biochemical measures, like BMI, WHR, and WHtR, are repeated in multiple sections. Consolidating these would make the results clearer and easier for readers to follow.

Point 8: The findings on sleep duration, shift type, and metabolic markers are interesting and important. I recommend to including exact p values and effect sizes throughout; this would make the results more transparent and credible.

Point9: The associations between short or interrupted sleep and BMI, blood pressure, and fasting glucose are compelling. It would strengthen the results presentation to clarify whether these analyses accounted for potential confounders like age, sex, or workload.

Pont 10: In the discusion section, the observation that short sleep was associated with lower BMI is intriguing. It may be helpful to clarify that this could reflect short-term effects (e.g., increased energy expenditure during extended wakefulness) rather than long-term metabolic outcomes, to avoid misinterpretation.

Point 11: I suggest explicitly highlighting the novelty of this study in the conclusion. Emphasizing that this is the first study in Ethiopia examining MS among healthcare professionals working long shifts would reinforce its relevance and potential impact on occupational health policies and targeted interventions.

Point 12: I understand the reliance on self-reported behavioral and lifestyle data, but it would be helpful to explicitly acknowledge the possibility of social desirability bias, which may affect accuracy. Additionally, some smaller subgroups, like night-shift workers, might limit generalizability. You might suggest that future studies consider objective monitoring methods or larger sample sizes to strengthen the findings.

Reviewer #2: Refer to the comments in the document

6. PLOS authors have the option to publish the peer review history of their article (what does this mean?). If published, this will include your full peer review and any attached files.

Reviewer #1: No

Reviewer #2: No

---

## [Author Response · Author response to Decision Letter 1]

4 Apr 2026

Response to Reviewers

Dear editor

We thank you very much for your constructive comments and suggestions for the improvement of this manuscript. We have corrected and amended the manuscript based on your suggestions. To visualize the changes made in the manuscript, we enabled the track change feature in Microsoft word.

Editorial comments

Comment 1: Please ensure that your manuscript meets PLOS ONE's style requirements, including those for file naming.

Response: - Thank you for this important comment. We carefully revised the manuscript to fully comply with PLOS ONE formatting and style requirements.

Comment 2: Your ethics statement should only appear in the Methods section of your manuscript. If your ethics statement is written in any section besides the Methods, please delete it from any other section.

Response: Thank you for this valuable comment. We have revised the manuscript accordingly. The ethics statement has been retained only in the Methodology section (page 11, lines 278–288) and has been removed from the Declarations section to ensure compliance with PLOS ONE’s formatting requirements.

Comment 3: All PLOS journals require all data underlying the findings to be freely available either in a public repository, within the manuscript, or as supplementary information.

Response: Thank you for this important comment. We have revised the Data Availability Statement to comply with PLOS ONE’s policy. All relevant data underlying the findings are now provided within the manuscript and its Supporting Information files. The complete de-identified dataset has been uploaded as S1 Dataset in the supplementary files.

Comment 4: Please include captions for your Supporting Information files at the end of your manuscript, and update any in-text citations to match accordingly.

Response: Thank you for this helpful comment. We have added the Supporting information captions section at the end of the manuscript after the references. The caption for the uploaded supplementary dataset has been included as well.

Comment 5: If the reviewer comments include a recommendation to cite specific previously published works, please review and evaluate these publications to determine whether they are relevant and should be cited.

Response: Thank you for this guidance. We will carefully review all reviewer comments and evaluated the relevance of any suggested references.

Reviewer comments

Dear Reviewers, we sincerely thank you for the thoughtful and constructive feedback provided on our manuscript. We have carefully considered all comments and suggestions and have revised the manuscript accordingly. Below, we provide a detailed, point-by-point response to each comment. All changes made in the manuscript have been with track change for clarity.

Reviewer 1 comments

Reviewer Comment – Point 1: In the intro section, you mention NHANES data up to 2018, which is fine, but since newer reports exist, the introduction would benefit from including the latest trends, at least briefly. It shows the reader that you are fully up to date.

Response: Thank you for this insightful comment. We agree that incorporating the most recent trends strengthens the background section. Accordingly, we updated the Introduction by adding recent NHANES evidence through 2023. This revision has been incorporated in the Introduction section (Page 4, Line Number 61-67).

Reviewer Comment – Point 2: The shift from general mechanisms to the Ethiopian setting feels a bit abrupt. You might add one sentence explaining why low- and middle-income settings are particularly vulnerable (e.g., understaffing, resource constraints, high patient load).

Response: Thank you for this excellent suggestion. We agree that a clearer transition improves the logical flow of the Introduction. Accordingly, we added a bridging sentence highlighting the increased vulnerability of low- and middle-income healthcare settings such as African context which may exacerbate metabolic risk among healthcare professionals. This sentence has been inserted immediately before the Ethiopian context in the Introduction section (Page 5, Line Number 98-104).

Reviewer Comment – Point 3: Your description of the study setting and period is clear. However, you may want to justify why these three specific hospitals were selected and whether their patient/worker profile differs from other facilities in the region. This would help clarify representativeness and potential site-selection bias.

Response: Thank you for this valuable comment. We have clarified the rationale for selecting the three hospitals in the Methods section. Specifically, we mentioned that the hospitals were chosen in the first stage of a multistage random sampling procedure from all eligible primary hospitals in the Central Gondar Zone, ensuring that each facility had an equal probability of inclusion and minimizing site-selection bias. We also added a statement noting that the selected hospitals share comparable service and workforce characteristics with other primary hospitals in the region, thereby supporting the representativeness of the sample (Page 8, line numbers 177-179).

Reviewer Comment – Point 4: Some exclusion criteria (e.g., “serious conditions,” “recent major surgery”) are very broad and open to subjective interpretation. Please clarify how these were assessed (self-report, clinician confirmation, medical records).

Response: Thank you for this important comment. We have revised the Exclusion Criteria section to clarify that “recent major surgery” and “serious conditions” were identified through a self-report from participants, ensuring objective and reproducible assessment. We also specified a time frame for recent surgery (within the past six months) to reduce ambiguity (please see page 7, lines 140 - 142).

Reviewer Comment – Point 5: Anthropometry and blood pressure measurement require clarification: were tools calibrated daily, were measurements taken twice and averaged, and was the same arm used for BP?

Response: Thank you for this important comment. We have revised the Methods section to clarify that all anthropometric instruments (digital weighing scale and stadiometer) were calibrated before the actual measurement. All measurements were taken twice and averaged, with a third measurement taken if the first two readings differed beyond a pre-specified threshold. Blood pressure measurements were consistently taken on the non-dominant upper arm after a minimum of 5 minutes of rest. We believe these revisions ensure clarity, reproducibility, and adherence to standardized measurement procedures (please see page 11 - 12, lines 271 - 291).

Reviewer Comment – Point 6: If DDS was based on a single day, this is highly vulnerable to day-to-day variation. Clarify whether it was one recall or multiple days, and if only one day was used, it is very important to acknowledge this limitation.

Response: Thank you for this important comment. In our study, DDS was assessed based on participants’ habitual dietary intake, reflecting their routine eating patterns rather than a single 24-hour recall (page 9, lines 204 - 206). We have clarified this in the Methods section and also acknowledge that self-reported dietary data may be subject to recall bias in the methodology section, which could influence the accuracy of DDS assessment (page 21, lines 568–570).

Reviewer Comment – Point 7: The definition (“20–30 min daily”) is not aligned with international norms (e.g., WHO). This may misclassify participants and affect associations. You should justify or revise the operational definition.

Response: Thank you for this comment. We have revised the operational definition of physical activity in the Methods section to align with WHO 2020 guidelines, which recommend at least 150 minutes of moderate-intensity aerobic activity per week (~30 minutes per day on most days). Our frequency-based categories (Inactive, Low, Moderate, High) reflect this threshold while ensuring comparability and minimizing misclassification of participants’ activity levels (Page 10, lines 234–241).

Reviewer Comment – Point 8: It would be useful to specify how multicollinearity was assessed and handled, why p<0.25 was used for bivariable screening (cite reference), and how missing data were treated.

Response: Thank you for this comment. We have revised the Statistical Analysis section in methodology to clarify that Multicollinearity among independent variables was assessed since we have used STATA v. 14 for statistical analysis it automatically omits multicollinear variables however VIF and tolerance were checked and variables exceeding thresholds were excluded from multivariable analysis. whereas bivariable logistic regression used p < 0.25 as the screening threshold, following the standard recommendation to avoid omitting potentially important variables. And missing data were minimal (<1%) and handled using complete-case analysis. We believe these clarifications enhance transparency and reproducibility of our analysis (please see page 13, lines 320–330).

Reviewer Comment – Point 9: Some anthropometric and biochemical measures, like BMI, WHR, and WHtR, are repeated in multiple sections.

Response: Thank you for this comment. We went to clarified the presentation of anthropometric and some biochemical measures: Table 2 presents categorical distributions while Table 5 presents mean values of the same measures as outcome variables by metabolic syndrome status. In the text, we highlight key findings and refer to the respective tables, avoiding repetition of exact numbers.

Reviewer Comment – Point 10: The findings on sleep duration, shift type, and metabolic markers are interesting and important. I recommend including exact p-values and effect sizes throughout; this would make the results more transparent and credible.

Response: Thank you for this valuable suggestion. We agree that reporting exact p-values and effect sizes improves transparency and interpretability of the findings. Accordingly, we revised the results section and Table 6 to include exact p-values for all comparisons and appropriate effect size measures (Cohen’s d for independent t-tests and eta squared for one-way ANOVA). These additions strengthen the statistical reporting and improve the credibility of the reported associations (please see page 15, lines 385–400 and page 31, Page 33-34 Revised Table 6 ).

Reviewer Comment – Point 11: The associations between short or interrupted sleep and BMI, blood pressure, and fasting glucose are compelling. It would strengthen the results presentation to clarify whether these analyses accounted for potential confounders like age, sex, or workload.

Response: Thank you for this important observation. The comparisons of mean BMI, blood pressure, and fasting glucose across sleep duration and sleep interruption categories were conducted using independent t-tests and one-way ANOVA, and therefore represent unadjusted analyses. We have clarified this in the Results section. Potential confounding variables such as age, sex, and workload were accounted for separately in the multivariable logistic regression model, where factors independently associated with metabolic syndrome were assessed.

Reviewer Comment – Point 12: In the discussion section, the observation that short sleep was associated with lower BMI is intriguing. It may be helpful to clarify that this could reflect short-term effects (e.g., increased energy expenditure during extended wakefulness) rather than long-term metabolic outcomes, to avoid misinterpretation.

Response: Thank you for this insightful comment. We indeed agree that such finding requires careful interpretation. Accordingly, we revised the Discussion section to clarify that the observed lower BMI among participants with shorter sleep duration may reflect acute or short-term effects of prolonged wakefulness rather than a beneficial long-term metabolic effect. We also emphasized that chronic sleep deprivation remains associated with adverse metabolic outcomes in longitudinal evidence (please see page 18, lines 477–484).

Reviewer Comment – Point 13: I suggest explicitly highlighting the novelty of this study in the conclusion. Emphasizing that this is the first study in Ethiopia examining metabolic syndrome among healthcare professionals working long shifts would reinforce its relevance and potential impact on occupational health policies and targeted interventions.

Response: Thank you for this excellent suggestion. We have revised the Conclusion section to explicitly highlight the novelty of our study as the first Ethiopian study assessing metabolic syndrome among healthcare professionals working prolonged shifts. We also strengthened the policy relevance by emphasizing the implications for occupational health policies, targeted workplace interventions, sleep hygiene, and routine metabolic screening programs for healthcare workers (page 20, lines 528–531).

Reviewer Comment – Point 14: I understand the reliance on self-reported behavioral and lifestyle data, but it would be helpful to explicitly acknowledge the possibility of social desirability bias, which may affect accuracy. Additionally, some smaller subgroups, like night-shift workers, might limit generalizability. You might suggest that future studies consider objective monitoring methods or larger sample sizes to strengthen the findings.

Response: Thank you for this thoughtful suggestion. We already mentioned the effect of social desirability bias on this particular study and now we have expanded the limitations section to explicitly acknowledge the potential for social desirability bias and recall bias associated with self-reported behavioral and lifestyle variables. We also noted that the relatively smaller size of certain subgroups, particularly exclusive night-shift workers, may limit the generalizability of subgroup-specific findings. Moreover, we incorporated your suggestion for future studies to use objective monitoring approaches and larger multicenter samples to strengthen the robustness and external validity of future research (page 21, lines 572–577).

Reviewer 2 comments

Title

Reviewer 2 Comment 1: The title is quite long and could benefit from tightening for readability and engagement. Readers also need to know why employees working in the same institution may work different hours.

Response: Thank you for this valuable suggestion. We have revised the title to improve readability and conciseness while preserving the key study concepts. Our updated title read as “Prevalence and determinants of metabolic syndrome among long-shift healthcare professionals in primary hospitals of Central Gondar Zone, Northwest Ethiopia”; In addition, we added a clarifying sentence in the Methods section explaining that healthcare professionals within the same hospital may work different hours depending on their professional role, ward assignment, emergency duties, staffing shortages, and rotating schedules. This helps clarify the variability in shift duration exposure among workers in the same institution (page 5, lines 109–112).

Abstract

Reviewer 2 – Minor structural and organizational comments: Several suggestions were made regarding repositioning sentences, paragraphs, and tables to improve the logical flow of the manuscript.

Response: Thank you for these helpful editorial suggestions. We carefully revised the manuscript structure by relocating the indicated sentences, paragraphs, and table descriptions to their more appropriate sections as recommended. These changes improved the logical progression of the Introduction, Methods, Results, and Discussion sections and enhanced overall readability.

Reviewer Comment: The recommendation in the abstract seems to be overgeneralized. Narrow to hospital caregivers/workplace group.

Response: Thank you for this valuable suggestion. We have revised the abstract conclusion to make the recommendation more specific to the hospital workplace setting and healthcare professionals working prolonged shifts. The revised statement now emphasizes routine metabolic screening, sleep hygiene support, and improved shift scheduling targeted specifically to hospital staff and high-risk worker groups (page 2-3, lines 44–48).

Introduction

Reviewer Comment: Suggested paraphrasing of the sentence regarding

---

## [Decision Letter · Decision Letter 1]

19 May 2026

Prevalence and determinants of metabolic syndrome among long-shift healthcare professionals in primary hospitals of Central Gondar Zone, Northwest Ethiopia

PONE-D-25-35581R1

Dear Dr. Chane,

We’re pleased to inform you that your manuscript has been judged scientifically suitable for publication and will be formally accepted for publication once it meets all outstanding technical requirements.

Kind regards,

Edmund Tetteh Nartey

Academic Editor

PLOS One

Additional Editor Comments (optional):

Reviewers' comments:

Reviewer's Responses to Questions

**Comments to the Author**

1. If the authors have adequately addressed your comments raised in a previous round of review and you feel that this manuscript is now acceptable for publication, you may indicate that here to bypass the “Comments to the Author” section, enter your conflict of interest statement in the “Confidential to Editor” section, and submit your "Accept" recommendation.

Reviewer #1: All comments have been addressed

2. Is the manuscript technically sound, and do the data support the conclusions?

Reviewer #1: Yes

3. Has the statistical analysis been performed appropriately and rigorously? 

Reviewer #1: Yes

4. Have the authors made all data underlying the findings in their manuscript fully available?

Reviewer #1: Yes

5. Is the manuscript presented in an intelligible fashion and written in standard English?

Reviewer #1: Yes

6. Review Comments to the Author

Reviewer #1: Dear Authors,

Thank you for your efforts in revising the manuscript and for the detailed responses to the reviewers’ comments. The authors have adequately addressed the concerns raised during peer review and implemented substantial revisions throughout the manuscript. The revised version provides clearer methodological descriptions, improved statistical reporting, and a more balanced interpretation of the findings, particularly in the discussion and limitations sections. Overall, the revisions have strengthened the clarity, transparency, and scientific rigor of the study. I therefore consider the manuscript acceptable for publication in its revised form

7. PLOS authors have the option to publish the peer review history of their article (what does this mean?). If published, this will include your full peer review and any attached files.

Reviewer #1: No

---

## [Editor Report · Acceptance letter]

PONE-D-25-35581R1

PLOS One

Dear Dr. Chane,

I'm pleased to inform you that your manuscript has been deemed suitable for publication in PLOS One. Congratulations! Your manuscript is now being handed over to our production team.

Kind regards,

on behalf of

Dr. Edmund Tetteh Nartey

Academic Editor

PLOS One